# Research

materials science

composites, alumina shapes, dielectric properties,
UV-cured epoxy acrylic

**Authors for correspondence:**
Qinghao Yang
e-mail: yangxjtu@hotmail.com
Guanjun Zhang
e-mail: gjzhang@xjtu.edu.cn

This article has been edited by the Royal Society of Chemistry, including the commissioning, peer review process and editorial aspects up to the point of acceptance.

# Effect of alumina shapes on dielectric properties of UV-cured epoxy acrylic composite with alumina

Jiepeng Bian[1], Qiuli Zhao[1], Zhenzhong Hou[1], Jie Dong[1], Qinghao Yang[1] and Guanjun Zhang[2]

[1]College of Materials Science and Engineering, Xi'an University of Science and Technology, Xi'an 710054, People's Republic of China
[2]State Key Laboratory of Electrical Insulation and Power Equipment, School of Electrical Engineering, Xi'an Jiaotong University, Xi'an 710049, People's Republic of China

JB, 0000-0002-3775-5453; QY, 0000-0002-1384-2982;
GZ, 0000-0003-1859-0443

Polymer-based composites with the advantages of ceramics and polymers exhibit high dielectric constant, good processing properties and low dielectric loss. The composites with a varied content of irregular alumina ($i$-$Al_2O_3$) filler were prepared by UV-cured epoxy acrylic (EA). Spherical alumina ($s$-$Al_2O_3$) was used as a filler to further investigate the effect of alumina ($Al_2O_3$) shapes on dielectric properties of composites in the frequency range of 50 Hz–1 MHz. Fourier transform infrared spectroscopy proved that the UV-cured epoxy acrylic/alumina ($Al_2O_3$/EA) composites were successfully fabricated. Scanning electron microscopy demonstrated that $i$-$Al_2O_3$ particles have superior homodispersion in the matrix. Through testing, for all samples, with the addition of $Al_2O_3$, the relative permittivity of composites increased as expected, and the dielectric loss decreased accordingly. These data show that the incorporation of $i$-$Al_2O_3$ particles presents better properties when compared with $s$-$Al_2O_3$/EA, which indicates that $i$-$Al_2O_3$ particles have more influence on the dielectric properties of the composites than those of $s$-$Al_2O_3$ particles. According to Weibull distribution, the characteristic breakdown strength of the $Al_2O_3$/EA composites was obtained and the results suggested that the composites of $i$-$Al_2O_3$/EA exhibited better breakdown performance.

# 1. Introduction

Polymer-based insulation materials with high dielectric constant have been widely applied in the electrical and electronic field,

especially in capacitors, motors and cables [1–3]. However, polymer materials generally exhibit low dielectric constant, and the energy density cannot be greatly improved [4], which limits the application of polymer materials in insulation field. Generally, one approach to enhance the relative permittivity of polymer is to incorporate the particles with a higher level of permittivity as fillers, such as inorganic ceramic powders (e.g. $BaTiO_3$ and $Ba_xSr_{1-x}TiO_3$) [5–7] and particles with high electric conductivity like metal and carbon nanotubes [8–10]. Recently, Ghosh et al. [11] reported a composite film with polyvinylidene fluoride (PVDF) as matrix, and the silver (Ag) nanoparticles decorated $CaCu_3Ti_4O_{12}$ (CCTO) as a filler showed high dielectric constant, which is about 20% higher than that of pure CCTO. He et al. [12] prepared a core−shell structure nanoparticle and used it as a filler to fabricate $BT@Al_2O_3$/PVDF composite films, which show a low dielectric loss (about 0.02 in 1 kHz) with high dielectric constant. Ceramic powders, metal particles and carbon nanotubes are incorporated in the polymer in an appropriate proportion, which can bring the advantages of both in composite materials [13–16]. Therefore, the preparation of polymer-based dielectric composites has become a hotspot and focus in engineering dielectric materials.

Alumina is commonly applied as a heat conduction material [17,18]. Due to the good thermal conductivity of alumina, in recent years, many scholars have conducted research on the effect of alumina on the dielectric properties of materials. Sudha et al. [19] reported on nanostructured amorphous alumina-modified polycarbonate composites, which showed the enhancement of dielectric constant over five times of pure polycarbonate and the loss decreased by 13.3% with 5 wt % $Al_2O_3$ as a filler and suggested that the composites could serve as a better electrical insulator in capacitors, printed circuit boards (PCBs) and electronic packaging industries [20]. Wang et al. [21] prepared a composite of $Al_2O_3$/LDPE (low density polyethylene) and showed that nano-$Al_2O_3$ could increase the electrical strength and electrical resistance of LDPE partly. It has been proved that $Al_2O_3$ as a functional filler can enhance the dielectric properties of inner insulation materials. It was found that with $Al_2O_3$ loading in composite materials, the relative permittivity increases, while the dielectric loss decreases; moreover, the breakdown strength will reduce accordingly [22,23]. However, nanoparticles tend to agglomerate in the polymer, which leads to increased dielectric loss and lower breakdown strength of the composites. Besides, at present, most investigators have focused on the spherical nano-alumina particles, and few have studied micro-$Al_2O_3$ particles. Moreover, the effects of $Al_2O_3$ particle shapes on dielectric properties of the composites were rarely reported.

Currently, a great majority of the polymer-based dielectric composites are formed by hot-pressing. This method is complicated, energy-intensive and requires a large amount of organic solvents, which is unfriendly to the environment [24,25]. Furthermore, evaporating the solvent during the curing process will make the surface of materials rough. In this work, the UV-curing technology was adopted to fabricate composite materials, due to the UV-curing technology being a high-efficiency and energy-saving, no evaporative volatile solvents and high-quality material surface treatment method that has been widely used in printing, electronic packaging and aerospace, etc. [26,27]. With the advancement of technology, a Japanese scholar has combined photocuring and 3D printing to print the sheet samples of alumina/UV-cured acrylic composites [23], which confirm the feasibility of UV-curing technology. Epoxy acrylic is the photocuring oligomer with the fast curing rate, which has excellent chemical resistance, good thermal stability and electrical properties. Epoxy acrylic is also the most widely used resin on the market [28–31]. Based on this, in this paper, the bisphenol A epoxy acrylic, diluent and photoinitiator were mixed in proper proportions to form the polymer matrix, two shapes of alumina particles sourced commercially were incorporated to enhance the relative permittivity of polymer materials, and cured by exposure to UV light. The dispersity of alumina particles in the matrix was observed by scanning electron microscopy (SEM), and the effect of alumina shapes on the dielectric properties of composites was investigated.

# 2. Experimental

## 2.1. Materials

Bisphenol A epoxy acrylic (EA) was provided by Nanjing Jiazhong Chemical Technology, China. Hexanediol diacrylate (HDDA, 99%) was used as a diluent to adjust the viscosity of the system, which was supplied by Henan Tianfu Chemical and used as received. A photoinitiator named 1-hydroxycycloethyl phenyl ketone (UV184, 98%) was purchased from Alfa Aesar and used as received. Irregular alumina (i-$Al_2O_3$) with a particle size of about 10–18 μm was provided by Taian Shengyuan Powder, China. Spherical alumina (s-$Al_2O_3$) with an average particle size of 15 μm was obtained from Ya'an Bestry Advanced Materials Co., Ltd, China.

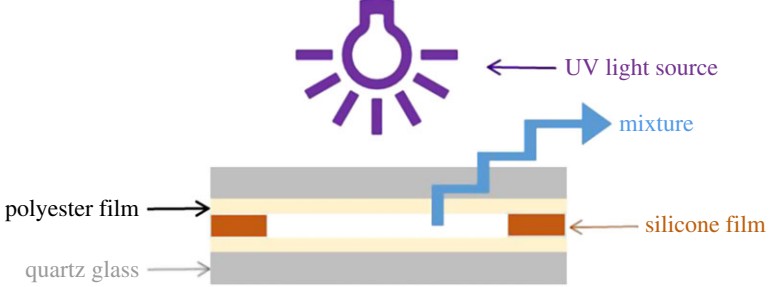

**Figure 1.** Diagrammatic sketch of UV-cured composites.

## 2.2. Preparation for the Al$_2$O$_3$/EA composites

The Al$_2$O$_3$/EA composites were prepared by UV-curing. The specific preparation process is as follows: to form a matrix, bisphenol A epoxy acrylic (EA), HDDA and UV184 were mixed at a mass ratio of 20 : 4 : 1. Then, the irregular or spherical alumina was added to a polymer matrix as a filler with different volume content (10%, 20%, 30%, 40% and 50%). After mechanical stirring for 1 h, the homogeneous mixture was placed in a vacuum to remove air bubbles for 30 min at room temperature. Finally, the mixture was poured into the mould, the sheet samples with an average thickness of 0.2 mm were prepared by UV light irradiating the mixture for 20 s and the diagrammatic sketch of UV light curing is shown in figure 1. An unfilled resin sample was prepared as a control sample according to the above procedure.

## 2.3. Characterizations

To confirm whether the Al$_2$O$_3$/EA composites were cured completely, the structures of the composites were investigated by Fourier transform infrared spectroscopy (FTIR, Nicolet Impact-400, USA). The dielectric properties of the composite materials were dominated by the interfacial bonding between the matrix and the filler, and the homogeneous dispersity of filler particles in the matrix. The dispersion of the Al$_2$O$_3$ particles in the matrix and the interface state of the two phases were observed by a scanning electron microscope (SEM, SEC4500M, Korea) at a voltage of 10 kV, and samples were sprayed gold before testing. The dielectric properties of the composites were analysed by measuring the capacitance and dielectric loss tangent of the samples with an impedance analyser (Agilent E4980A, USA) from 50 Hz to 1 MHz under the measuring voltage of 1.0 V. A layer of conductive silver paste was placed on both sides of the sample before testing. The direct-current breakdown strength of composites was estimated on an LCR meter (YD-5750, Xinyuan Electric, China) with the voltage rise rate of 0.4 kV s$^{-1}$. The test system for direct-current breakdown strength adopts ball electrodes with a diameter of 25 mm, which were placed in the transformer oil at 33°C to prevent surface flashover during the test.

# 3. Results and discussion

## 3.1. FTIR analysis

The chemical structures of the uncured EA and the cured Al$_2$O$_3$/EA composites were characterized by FTIR spectroscopy, as depicted in figure 2. In the spectra of uncured resin, it can be seen that the absorption peak of C=O bonds was at 1720 cm$^{-1}$, the characteristic peak of C=C bonds in epoxy acrylic oligomers was at 1637 cm$^{-1}$, the skeletal vibration of the para-substituted benzene ring was at 1508 cm$^{-1}$ and the C–O–C stretching vibration of fat aromatic ether was at 1262 cm$^{-1}$. In addition, there are no characteristic peaks at 1066 and 590 cm$^{-1}$ in the uncured resin, which are the bending vibrations of the OH and Al–O bonds, respectively, presented in Al$_2$O$_3$/EA composites. According to the UV-curing mechanism, under the effect of photoinitiator, the C=C bonds in functional monomers participate in the reaction to form a cross-linked network structure under the UV irradiation [27]. Obviously, there was not an absorption peak of C=C bonds in the cured samples. In other words, there was no acrylic oligomer in cured Al$_2$O$_3$/EA composites, which indicates that the Al$_2$O$_3$/EA composites have been fabricated successfully.

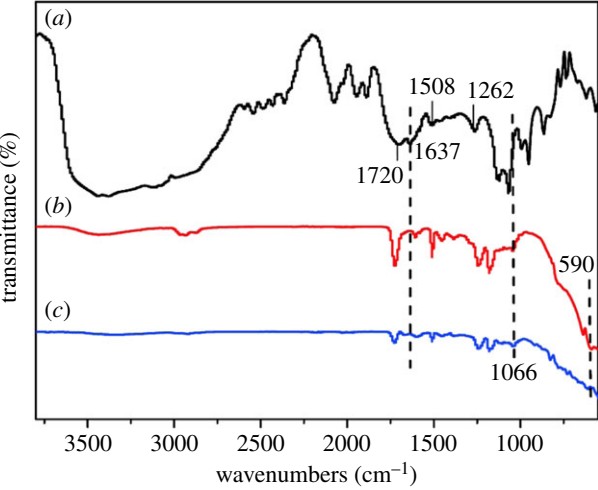

**Figure 2.** FTIR spectra of (*a*) uncured EA and cured composites, (*b*) *i*-Al$_2$O$_3$/EA and (*c*) *s*-Al$_2$O$_3$/EA.

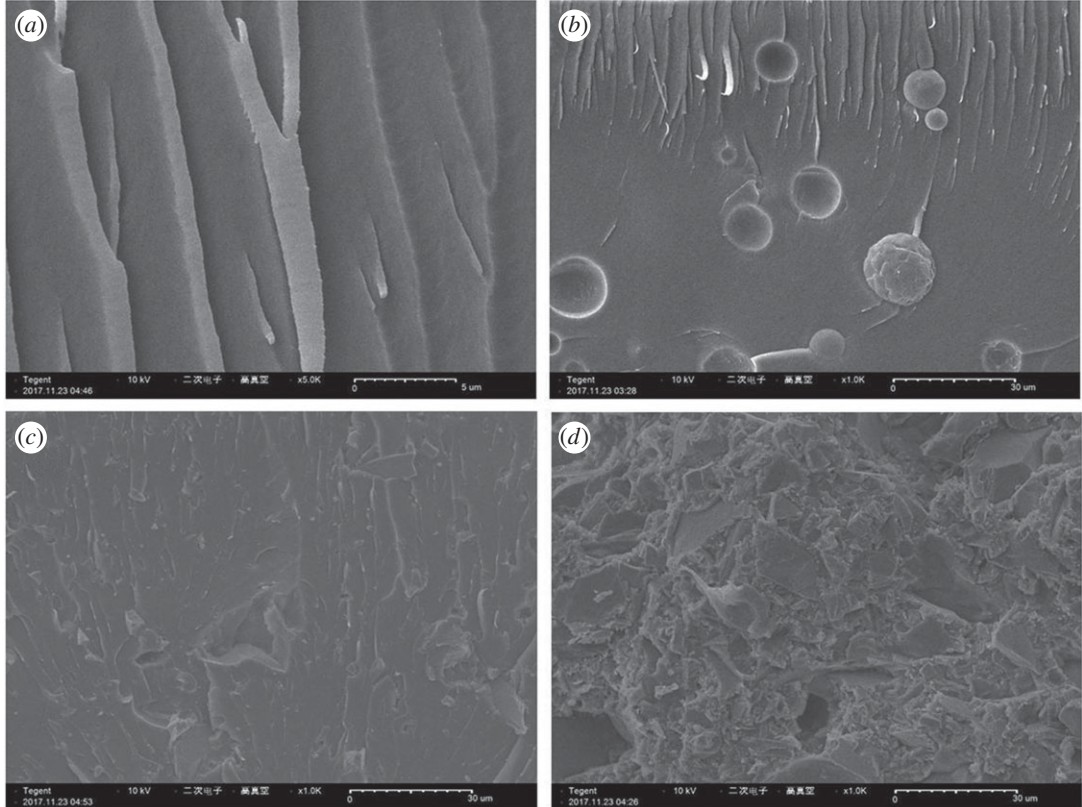

**Figure 3.** SEM of samples: (*a*) unfilled resin, (*b*) 10 vol% *s*-Al$_2$O$_3$, (*c*) 10 vol% *i*-Al$_2$O$_3$ and (*d*) 50 vol% *i*-Al$_2$O$_3$.

## 3.2. Microstructure of composites

Figure 3 shows the SEM images of unfilled resin and Al$_2$O$_3$/EA composite samples. As seen from figure 3*b*, evidently non-uniform dispersion and deposition of *s*-Al$_2$O$_3$ particles can be observed in the resin, and almost a pure resin layer in the upper layer of the composite sample at the filler volume fraction of 10%, which is attributed to the fact that the *s*-Al$_2$O$_3$ particles have stronger flowability in the resin than the irregular ones, especially less filler content resulting in low viscosity of the system. Meanwhile, in figure 3*c*, the *i*-Al$_2$O$_3$ particles bond well to the matrix in the sample with *i*-Al$_2$O$_3$ of 10 vol% and present a homogeneous dispersity in the matrix. Significantly, increasing defects appeared as the filler loading increased, which might be because the interface is increased of Al$_2$O$_3$ particles and the air bubbles are hard to remove completely in the mixture with a high viscosity.

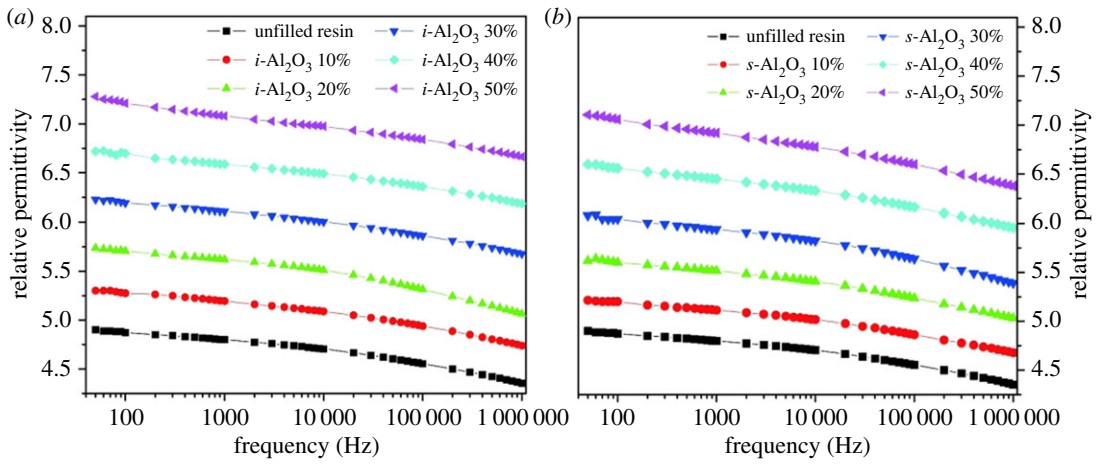

**Figure 4.** Frequency dependence of relative permittivity of composites: (*a*) *i*-Al₂O₃/EA and (*b*) *s*-Al₂O₃/EA with varied content measured at room temperature.

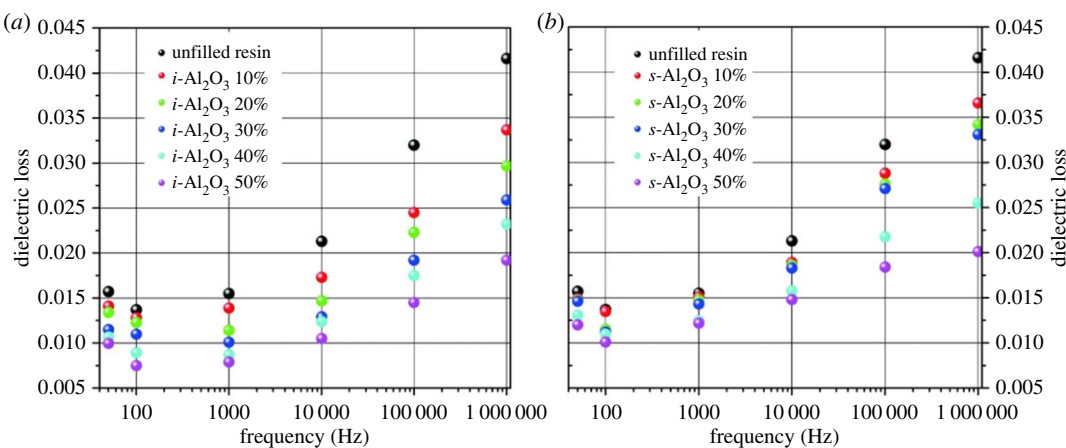

**Figure 5.** Frequency dependence of dielectric loss of composites: (*a*) *i*-Al₂O₃/EA and (*b*) *s*-Al₂O₃/EA with varied content measured at room temperature.

## 3.3. Dielectric properties

To characterize the dielectric properties of materials, the capacitance and dielectric loss tangent were measured. According to the formula about capacitance and relative permittivity, the frequency dependence of relative permittivity of all prepared samples at room temperature is shown in figure 4. It can be observed that the relative permittivity of all samples decreases with increasing frequency, which is attributed to the dielectric relaxation phenomena of materials. The permittivity of composites was dominated by the interfacial polarization. With the frequency increasing, the interfacial polarization cannot keep up with the change of frequency, resulting in the relative permittivity decreasing correspondingly [32]. Moreover, the curves illustrated that by increasing the content of Al₂O₃, the relative permittivity increased as expected and was all higher than that of pure resin [23,33–36]. Notably, the *i*-Al₂O₃/EA composites show higher relative permittivity compared with *s*-Al₂O₃/EA composites at the same Al₂O₃ concentration. It can be seen that the relative permittivity for *i*-Al₂O₃/EA composites increases from 5.30 to 7.28 as the volume fraction of filler incorporation is increased from 10% to 50% and 5.21 to 7.10 for *s*-Al₂O₃/EA at 50 Hz; however, the relative permittivity for unfilled resin is only 4.90.

Figure 5 is a graph of dielectric loss of the resulted samples. As the dielectric loss of Al₂O₃ is lower than that of the resin [23], decreased dielectric loss was obtained as Al₂O₃ content increased. As seen from figure 5, the *i*-Al₂O₃/EA composites present lower dielectric loss compared with that of *s*-Al₂O₃/EA composites at the same content. The dielectric loss of *i*-Al₂O₃/EA is decreased from 0.0157 to 0.0099 and the dielectric loss of *s*-Al₂O₃/EA from 0.0157 to 0.012, which is mainly attributed to more defects that appeared in samples with *s*-Al₂O₃ resulting in a relatively higher loss.

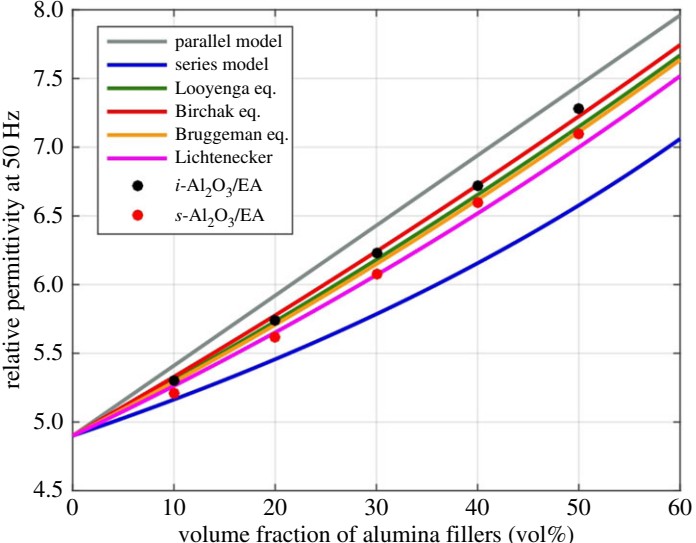

**Figure 6.** Dependence of relative permittivity on the filler volume fraction in composites at 50 Hz.

Obviously, $i$-Al$_2$O$_3$/EA composites show better dielectric properties than the $s$-Al$_2$O$_3$/EA ones, which is due to the fact that spherical particles were prone to aggregate and deposit, resulting in more defects formed between the filler and matrix.

Figure 6 shows the dependence of relative permittivity on the volume fraction of alumina at 50 Hz. To verify the rationality of the experimental data, we compared a series of theoretical models [37]. In the figure, the experimental data of all the prepared materials are between the permittivity values calculated by the parallel capacitor model and the series capacitor model, which correspond to the maximum and minimum values of two media composites. Besides, the experimental data are accordant to the Bruggeman's model and the Lichtenecker–logarithmic model. Furthermore, to study the effect of filler shape on the permittivity of two media composites and better fit the experimental data, we introduced a power law model [38,39]. A power low model is often used in dielectric modelling of the composite system, and the effective permittivity of a two-component composite can be modelled by the following equation:

$$\varepsilon_{\text{eff}}^{\beta_1} = \phi_1 \, \varepsilon_1^{\beta_1} + (1 - \phi_1) \, \varepsilon_2^{\beta_1} \, , \tag{3.1}$$

where $\varepsilon_{\text{eff}}$, $\varepsilon_1$ and $\varepsilon_2$ are the relative permittivity of the composite material, the filler and the matrix, respectively, $\Phi_1$ is the volume fraction of the filler and $\beta_1$ is the parameter of the shape and orientation of the filler within the composite. Here, the interface volume is not considered because the filler has micrometre-sized particles and its filling fraction is high. Looyenga [40] defines that $\beta_1$ value is $1/3$ for the spherical filler particles. We can see that the relative permittivity of the $s$-Al$_2$O$_3$/EA composite is closer to the Looyenga formula theoretical value. However, because of the incorporation of $i$-Al$_2$O$_3$, the relative permittivity tends to be in accordance with the Birchak formula [41], where $\beta_1$ value is $1/2$. The results show that the shape of the filler in composites has an important effect on the effective relative permittivity, and the larger the $\beta_1$ value, the higher the effective relative permittivity of the composite [39].

## 3.4. Breakdown strength

Breakdown strength is a critical parameter for evaluating the properties of dielectric composite materials. According to Weibull distribution (IEEE Standard 930-2004), data were processed; $\alpha$ and $\beta_2$ can be calculated by the following equations:

$$P_i = 1 - \exp\left[-\left(\frac{E}{\alpha}\right)^{\beta_2}\right], \tag{3.2}$$

where $E$, $\alpha$ and $\beta_2$ are the experimental value, the characteristic breakdown strength and the shape parameter about filler dispersion, respectively. $P_i$ is the cumulative probability of dielectric

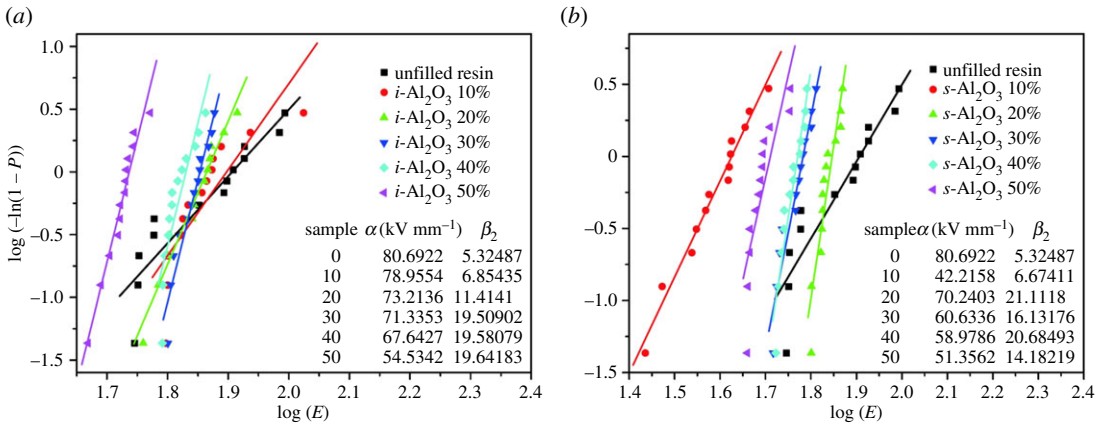

**Figure 7.** Two-parameter Weibull distribution of breakdown strength for the unfilled resin and composites with varied $Al_2O_3$ concentration: (a) $i$-$Al_2O_3$/EA and (b) $s$-$Al_2O_3$/EA.

breakdown, which should be calculated by the following formula as the number of test samples is less than 25:

$$P_i = \frac{i - 0.44}{n + 0.25},$$

(3.3)

The values of $\alpha$ and $\beta_2$ can be obtained after the linear fitting, as shown in figure 7. The introduction of $Al_2O_3$ particles into the resin reduced the $\alpha$ value of EA [42], which is caused by interfacial defects and air bubbles. It can be seen from figure 3, with increasing alumina content, more defects are found inside the materials, which results in the increased charge accumulation and partial discharge inside the materials. Comparatively, the $i$-$Al_2O_3$/EA composites show relatively higher $\alpha$ than that of $s$-$Al_2O_3$/EA composites, because the spherical particles are easier to aggregate and more defects are found inside the composites. Besides, the shape parameter $\beta_2$ of all $Al_2O_3$/EA composites is higher than that of unfilled resin, which reveals that the introduction of $Al_2O_3$ particles changes the breakdown mechanisms of EA. However, $\alpha$-value decreased greatly at the filling amount of 10 vol%, which is a result of heavy defects in the sample.

# 4. Conclusion

Composite materials were prepared by UV-curing successfully, which is handy and energy-saving. The microstructure of composites illustrated that $i$-$Al_2O_3$ particles disperse homogeneously and the $s$-$Al_2O_3$ particles tend to aggregate and deposit in the resin. With the increasing alumina content, the relative permittivity of $Al_2O_3$/EA composites increased, the dielectric loss decreased accordingly. However, the incorporation of $i$-$Al_2O_3$ particles presents better dielectric properties when compared with $s$-$Al_2O_3$/EA. The relative permittivity of $i$-$Al_2O_3$/EA composites with the filler volume fraction of 50% is up to 7.28 and the dielectric loss is only 0.099 at 50 Hz. The comparison of experimental data with the models of permittivity of two media composite indicates that the experimental data are reasonable. Based on the power law model, the parameter of the shape of the $i$-$Al_2O_3$ in the $i$-$Al_2O_3$/EA composite is 1/2, which is higher than that of $s$-$Al_2O_3$ in the $s$-$Al_2O_3$/EA composites, indicating that $i$-$Al_2O_3$ has a more important effect on the permittivity of the composite than that of $s$-$Al_2O_3$. Moreover, the $i$-$Al_2O_3$/EA composites exhibit better breakdown performance than the $s$-$Al_2O_3$/EA composites. Such UV-curing moulding method is expected to be used in preparing more insulation materials of excellent performance combined with 3D printers.

Data accessibility. All data and research materials supporting the results are in the article.

Authors' contributions. J.B. carried out the data analysis and drafted the manuscript. Q.Z. coordinated the study and helped draft the manuscript. Z.H. and J.D. participated in a statistical analysis and helped draft the paper. Q.Y. and G.Z. designed the study and carried out the statistical analyses. All authors gave final approval for publication.

Competing interests. We have no competing interests.

Funding. This work was financially supported by the National Natural Science Foundation of China (21204072 and U1766218), Natural Science Basic Research Plan in Shaanxi Province of China (2016JQ5078), the Key R&D Program

of Shaanxi Province, China (2017GY-133) and the Special Research Project of Shanxi Education Department (17JK0512).

Acknowledgements. We thank our anonymous reviewers for their insightful and helpful comments.

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
