## [Reviewer comments · Royal Society Open Science]

Review History

RSOS-180831.R0 (Original submission)

Review form: Reviewer 1

Is the manuscript scientifically sound in its present form?

No

Are the interpretations and conclusions justified by the results?

No

Is the language acceptable?

No

Is it clear how to access all supporting data?

Yes

Do you have any ethical concerns with this paper?

No

Have you any concerns about statistical analyses in this paper?

No

Recommendation?

Reject

Comments to the Author(s)

The paper presents an interesting method to improve dielectric properties of composites based on polymers and ceramics which can be used in the electronic industry. The following changes must be made in the article:

The wording of the abstract should be revised.

The word "Ahout" does not exist.

The word "compomsites" does not exist.

The word "expermental" does not exist.

Units and quantities must be separated.

Technical variables must be emphasized using italic fonts.

In FTIR analysis the presence of alumina must be emphasized.

In dielectric properties a more exhaustive list of papers where different models for determining electrical permittivity or electrical conductivity must be included by example "Analysis of DC Electrical Conductivity

Models of Carbon Nanotube-Polymer Composites with Potential Application to Nanometric Electronic Devices",

Journal of Electrical and Computer Engineering, Vol. 2013, Paper ID 179538, 14 pages, March 2013.

ISSN 2090-0155. In addition, results and discussion must be completely justified using other publications.

The contribution of the alumina in the composite must be completely justified using other publications.

The format used for the references does not correspond to the one required by the journal.

Figure 6 does not have labels on both axes.

The labels of the axes of Figure 7 should use italics for the technical variables.

Reference 1 is incomplete (authors) and incorrect (abbreviation of the journal).

Reference 2 has incomplete names of authors.

Reference 3 has incomplete names of authors.

Reference 5 has incomplete names of authors.

Reference 6 has incomplete names of authors.

Reference 15 must use subindices in chemical formulas.

Reference 20 has incomplete names of authors.

Reference 22 has incorrect names of authors.

Reference 24 must use subindices in chemical formulas.

Reference 25 must use subindices in chemical formulas.

Reference 26 has incorrect names of authors.

Reference 27 has incomplete and incorrect names of authors.

Reference 32 has incorrect names of authors.

Reference 36 has incorrect names of authors.

Review form: Reviewer 2

Is the manuscript scientifically sound in its present form?

Yes

Are the interpretations and conclusions justified by the results?

Yes

Is the language acceptable?

Yes

Is it clear how to access all supporting data?

Yes

Do you have any ethical concerns with this paper?

No

Have you any concerns about statistical analyses in this paper?

No

Recommendation?

Accept as is

Comments to the Author(s)

The manuscript "Effect of Alumina Shapes on Dielectric Properties of UVcured Epoxy Acrylic Composite with Alumina" is devoted to study of changes of dielectric properties after UV curing of aluminum oxide filled polymer matrix. The manuscript is well organized, the results are interesting and publishable. I recommend to accept the manuscript for publication.

Decision letter (RSOS-180831.R0)

15-Aug-2018

Dear Miss Bian:

Manuscript ID: RSOS-180831

Title: "Effect of Alumina Shapes on Dielectric Properties of UV-cured Epoxy Acrylic Composite with Alumina"

Thank you for submitting the above manuscript to Royal Society Open Science. Your paper was sent to reviewers and their comments are included at the bottom of this letter.

In view of the concerns raised by the reviewers, the manuscript has been rejected in its current form. However, a new manuscript may be submitted which takes into consideration these comments.

Please note that resubmitting your manuscript does not guarantee eventual acceptance, and that your resubmission will be subject to peer review before a decision is made.

Your resubmitted manuscript should be submitted by 12-Feb-2019. If you are unable to submit by this date please contact the Editorial Office.

Yours sincerely,
Dr Laura Smith, MRSC
Publishing Editor, Journals
Royal Society of Chemistry,
Thomas Graham House,
Science Park, Milton Road,
Cambridge, CB4 0WF, UK

Royal Society Open Science - Chemistry Editorial Office

On behalf of the Subject Editor Professor Anthony Stace and the Associate Editor Professor Claire Carmalt

REVIEWER(S) REPORTS:

Associate Editor Comments to Author ():

RSC Associate Editor:

Comments to the Author:

(There are no comments.)

RSC Subject Editor:

Comments to the Author:

(There are no comments.)

Reviewers' Comments to Author:

Reviewer: 1

Comments to the Author(s)

The paper presents an interesting method to improve dielectric properties of composites based on polymers and ceramics which can be used in the electronic industry. The following changes must be made in the article:

The wording of the abstract should be revised.

The word "Ahout" does not exist.

The word "compomsites" does not exist.

The word "expermental" does not exist.

Units and quantities must be separated.

Technical variables must be emphasized using italic fonts.

In FTIR analysis the presence of alumina must be emphasized.

In dielectric properties a more exhaustive list of papers where different models for determining electrical permittivity or electrical conductivity must be included by example "Analysis of DC Electrical Conductivity

Models of Carbon Nanotube-Polymer Composites with Potential Application to Nanometric Electronic Devices",

Journal of Electrical and Computer Engineering, Vol. 2013, Paper ID 179538, 14 pages, March 2013.

ISSN 2090-0155. In addition, results and discussion must be completely justified using other publications.

The contribution of the alumina in the composite must be completely justified using other publications.

The format used for the references does not correspond to the one required by the journal.

Figure 6 does not have labels on both axes.

The labels of the axes of Figure 7 should use italics for the technical variables.

Reference 1 is incomplete (authors) and incorrect (abbreviation of the journal).

Reference 2 has incomplete names of authors.

Reference 3 has incomplete names of authors.

Reference 5 has incomplete names of authors.

Reference 6 has incomplete names of authors.

Reference 15 must use subindices in chemical formulas.

Reference 20 has incomplete names of authors.

Reference 22 has incorrect names of authors.

Reference 24 must use subindices in chemical formulas.

Reference 25 must use subindices in chemical formulas.

Reference 26 has incorrect names of authors.

Reference 27 has incomplete and incorrect names of authors.

Reference 32 has incorrect names of authors.

Reference 36 has incorrect names of authors.

Reviewer: 2

Comments to the Author(s)

The manuscript "Effect of Alumina Shapes on Dielectric Properties of UVcured Epoxy Acrylic Composite with Alumina" is devoted to study of changes of dielectric properties after UV curing of aluminum oxide filled polymer matrix. The manuscript is well organized, the results are interesting and publishable. I recommend to accept the manuscript for publication.

Author's Response to Decision Letter for (RSOS-180831.R0)

See Appendix A.

RSOS-181509.R0

Review form: Reviewer 1

Is the manuscript scientifically sound in its present form?

Yes

Are the interpretations and conclusions justified by the results?

Yes

Is the language acceptable?

Yes

Is it clear how to access all supporting data?

No

Do you have any ethical concerns with this paper?

Yes

Have you any concerns about statistical analyses in this paper?

No

Recommendation?

Accept with minor revision (please list in comments)

Comments to the Author(s)

The proposed article is interesting because the development of dielectric composite materials is important for many applications in the electrical sector. However, some changes must be made as indicated below.

The word "compomsites" does not exist.

The word "exhibite" does not exist.

In Reference 11, one of the names of the authors is incorrect.

In Reference 35, the names of the authors is incomplete.

In Reference 37, Article ID must be used.

Decision letter (RSOS-181509.R0)

09-Oct-2018

Dear Miss Bian:

Title: Effect of Alumina Shapes on Dielectric Properties of UV-cured Epoxy Acrylic Composite with Alumina

Manuscript ID: RSOS-181509

Thank you for submitting the above manuscript to Royal Society Open Science. On behalf of the Editors and the Royal Society of Chemistry, I am pleased to inform you that your manuscript will

be accepted for publication in Royal Society Open Science subject to minor revision in accordance with the referee suggestions. Please find the reviewers' comments at the end of this email.

The reviewers and handling editors have recommended publication, but also suggest some minor revisions to your manuscript. Therefore, I invite you to respond to the comments and revise your manuscript.

Please also include the following statements alongside the other end statements. As we cannot publish your manuscript without these end statements included, if you feel that a given heading is not relevant to your paper, please nevertheless include the heading and explicitly state that it is not relevant to your work. We have included a screenshot example of the end statements for reference.

- Ethics statement

Please clarify whether you received ethical approval from a local ethics committee to carry out your study. If so please include details of this, including the name of the committee that gave consent in a Research Ethics section after your main text. Please also clarify whether you received informed consent for the participants to participate in the study and state this in your Research Ethics section.

OR

Please clarify whether you obtained the necessary licences and approvals from your institutional animal ethics committee before conducting your research. Please provide details of these licences and approvals in an Animal Ethics section after your main text.

OR

Please clarify whether you obtained the appropriate permissions and licences to conduct the fieldwork detailed in your study. Please provide details of these in your methods section.

Because the schedule for publication is very tight, it is a condition of publication that you submit the revised version of your manuscript before 18-Oct-2018. Please note that the revision deadline will expire at 00.00am on this date. If you do not think you will be able to meet this date please let me know immediately.

- 1) A text file of the manuscript (tex, txt, rtf, docx or doc), references, tables (including captions) and figure captions. Do not upload a PDF as your "Main Document".
- 2) A separate electronic file of each figure (EPS or print-quality PDF preferred (either format should be produced directly from original creation package), or original software format)
- 3) Included a 100 word media summary of your paper when requested at submission. Please ensure you have entered correct contact details (email, institution and telephone) in your user account

- 4) Included the raw data to support the claims made in your paper. You can either include your data as electronic supplementary material or upload to a repository and include the relevant doi within your manuscript
- 5) All supplementary materials accompanying an accepted article will be treated as in their final form. Note that the Royal Society will neither edit nor typeset supplementary material and it will be hosted as provided. Please ensure that the supplementary material includes the paper details where possible (authors, article title, journal name).

Best wishes,

Dr Laura Smith, MRSC
 Publishing Editor, Journals
 Royal Society of Chemistry,
 Thomas Graham House,
 Science Park, Milton Road,
 Cambridge, CB4 0WF, UK
 Royal Society Open Science - Chemistry Editorial Office

On behalf of the Subject Editor Professor Anthony Stace and the Associate Editor Professor Claire Carmalt.

RSC Associate Editor
 Comments to the Author:
 (There are no comments.)

Reviewer comments to Author:
 Reviewer: 1

Comments to the Author(s)
 The proposed article is interesting because the development of dielectric composite materials is important for many applications in the electrical sector. However, some changes must be made as indicated below.

The word "compomsites" does not exist.
 The word "exhibite" does not exist.
 In Reference 11, one of the names of the authors is incorrect.
 In Reference 35, the names of the authors is incomplete.

In Reference 37, Article ID must be used.

Author's Response to Decision Letter for (RSOS-181509.R0)

See Appendix B.

RSOS-181509.R1

Review form: Reviewer 1

Is the manuscript scientifically sound in its present form?

Yes

Are the interpretations and conclusions justified by the results?

Yes

Is the language acceptable?

Yes

Is it clear how to access all supporting data?

No

Do you have any ethical concerns with this paper?

No

Have you any concerns about statistical analyses in this paper?

No

Recommendation?

Accept as is

Comments to the Author(s)

Reference 37 should include Article ID 179538 because the data is incomplete.

Decision letter (RSOS-181509.R1)

23-Oct-2018

Dear Miss Bian:

Title: Effect of Alumina Shapes on Dielectric Properties of UV-cured Epoxy Acrylic Composite with Alumina

Manuscript ID: RSOS-181509.R1

It is a pleasure to accept your manuscript in its current form for publication in Royal Society Open Science. The chemistry content of Royal Society Open Science is published in collaboration with the Royal Society of Chemistry.

On behalf of the Subject Editor Professor Anthony Stace and the Associate Editor Professor Claire Carmalt.

RSC Associate Editor:
Comments to the Author:
(There are no comments.)

RSC Subject Editor:
Comments to the Author:
(There are no comments.)

Reviewer(s)' Comments to Author:
Reviewer: 1

Comments to the Author(s)
Reference 37 should include Article ID 179538 because the data is incomplete.

Appendix A

Response to Reviewer 1

We sincerely thank the reviewer for reviewing our manuscript and constructive suggestions. This reviewer raised a few questions about details. Following are our replies.

Question: Some corrections

We have carefully revised the misused words and expressions accordingly.

1. We revised the words: "about", "composites", "experimental".
2. We made all units and quantities separated.
3. We changed "I-Al₂O₃" and "S-Al₂O₃" to "i-Al₂O₃" and "s-Al₂O₃", which "i" and "s" are italic fonts.
4. We changed "α", "β₁" and "β₂" from non-italic fonts to italic fonts.
5. We analyzed the characteristic peaks of alumina in FTIR analysis. Please find the results and the description in the revised manuscript (Page 3).
6. We added the references "33-37" and "42" to further justify the rationality of the experimental results.

33 Wang Z, Zhou W, Sui X, Dong L, Cai H, Zuo J, Liu X, Chen Q. 2016 Dielectric studies of Al nanoparticle reinforced epoxy resin composites. *Polym. Composite*. 39, 887-894 (doi:10.1002/pc.24012)

34 Murudkar VV, Gaonkar AA, Deshpande VD, Mhaske ST. 2016 Comparison of dielectric properties of polydimethylsiloxane (PDMS) grafted polyacrylates/nano alumina and nano silica composites. *AIP Conf. Proc.* 1728, 020622(1-4). (doi:10.1063/1.4946673)

35 Rajesh S, Jantunen H. 2011 Low temperature sintering and dielectric properties of alumina-filled glass composites for LTCC applications. *Int. J. Appl. Ceram. Technol.* 9, 52-59. (doi:10.1111/j.1744-7402.2011.02684.x)

36 He S, Hu J, Zhang C, Wang J, Chen L, Bian X, Lin J, Du X. 2018 Performance improvement in nano-alumina filled silicone rubber composites by using vinyl trimethoxysilane. *Polym. Test.* 67, 295-301. (doi:10.1016/j.polymertesting.2018.03.023)

37 Vargas-Bernal R, Herrera-Pérez G, Calixto-Olalde ME, Tecpoyotl-Torres M. 2013 Analysis of DC electrical conductivity models of carbon nanotube-polymer composites with potential application to nanometric electronic devices. *Journal of Electrical and Computer Engineering*, 2013, 1-14. (doi:10.1155/2013/179538)

42 Wang SJ, Zha JW, Wu YH, Ren L, Dang ZM, Wu J. 2015 Preparation, microstructure and properties of polyethylene/alumina nanocomposites for HVDC insulation. *IEEE Trans. Dielectr. Electr. Insul.* 22, 3350-3356. (doi:10.1109/TDEI.2015.004903)

7. We added the labels of both axes in Figure 6.

8. We changed the labels of Figure 7 from non-italic fonts to italic fonts.

9. We revised the format used for the references including names of authors, abbreviation of the journal and chemical formulas.

Reference 1. We revised the authors' names from "Wang Y, Zhou X, Chen Q, Zhang Q" to "Wang Y, Zhou X, Chen Q, Chu B, Zhang Q" and the abbreviation of the journal to "IEEE Trans. Dielectr. Electr. Insul."

Reference 2. We revised the author's name from "Loye H" to "zur Loye HC".

Reference 3. We revised the author's name from "Zhang Q" to "Zhang QM".

Reference 5. We revised the authors' name from "Chan H L, Choy C, Wong K" to "Chan HLW, Choy CL, Wong KH".

Reference 6. We revised the authors' name from "Ge P, Tang X, Liu Q, Jiang Y, Li W, Li B" to "Ge PZ, Tang XG, Liu QX, Jiang YP, Li WH, Li B".

Reference 15. We changed chemical formulas to subindices.

Reference 19. We revised the author's name from "Sukumar R" to "Roy S".

Reference 20. We revised the author's name from "Montan G C" to "Montanari GC".

Reference 22. We revised the author's name from "Cho CD" to "Cho SD".

Reference 24. We changed chemical formulas to subindices.

Reference 25. We changed chemical formulas to subindices.

Reference 26. We revised the author's name from "Hua Y" to "Hu Y".

Reference 27. We revised the authors' name to "Park YJ, Lim DH, Kim HJ, Park DS, Sung IK".

Reference 32. We revised the authors' name from "Zhan J, Tian G, Wu Z, Qi S, Wu D" to "Zhan JY, Tian GF, Wu ZP, Qi SL, Wu DZ".

"Reference 36" was changed to "Reference 41". We revised the author's name from "Gardne CG" to "Gardner CG".

Response to Reviewer 2

This reviewer appreciated our work and recommended the publication of our paper. We sincerely thank him/her for careful reviewing our manuscript.

Appendix B

Dear editor

Thank you for your letter and for the reviewers' comments concerning our manuscript entitled "**Effect of alumina shapes on dielectric properties of UV-cured epoxy acrylic composite with alumina**" (RSOS-181509). We have made correction according to referee suggestions. The main correction in the paper and correspond to the reviewer's comments are as following:

Response to Reviewer 1

We sincerely thank the reviewer for reviewing our manuscript and constructive suggestions. This reviewer raised some minor revision. Following are our replies.

We have carefully revised the misused words and expressions accordingly.

1. We revised the words "compomsites" to "composites".
2. We revised the words "exhibite" to "exhibit".

We revised the references including names of authors and article ID:

Reference 11. We revised the author's name from "Tamayo Calderón RMT" to "Tamayo Calderón RM".

Reference 35. We revised the authors' name to "Rajesh S, Jantunen H, Letz M, Pichler-Willhelm S".

Reference 37. The Article ID is 179538, which is used in reference 37.

We have added the title of the moral statement:

Ethics. Our work is not relevant to animals. This item is not relevant to our work.

Besides, I want to explain that the corresponding authors of the article are Qinghao Yang and Guanjun Zhang.

I, represent all of authors, sincerely appreciate your responsible support.

Sincerely Yours,

Jiepeng Bian

2018.10.12